# Effect of Optical and Morphological Control of Single-Structured LEC Device

**DOI:** 10.3390/mi12070843

**Published:** 2021-07-19

**Authors:** Woo Jin Jeong, Jong Ik Lee, Hee Jung Kwak, Jae Min Jeon, Dong Yeol Shin, Moon Sung Kang, Jun Young Kim

**Affiliations:** 1Department of Semiconductor Engineering, Gyeongsang National University, 501 Jinjudaero, Jinju 52828, Korea; jeongwj95@gnu.ac.kr (W.J.J.); kwakhj@gnu.ac.kr (H.J.K.); jmjeon95@gnu.ac.kr (J.M.J.); dyshin@gnu.ac.kr (D.Y.S.); 2Department of Chemical and Biomolecular Engineering, Sogang University, Seoul 04107, Korea; airfoil@sogang.ac.kr (J.I.L.); kangms@sogang.ac.kr (M.S.K.); 3Institute of Emergent Materials, Sogang University, Seoul 04107, Korea

**Keywords:** light-emitting electrochemical cell, single-structure, thickness, heat treatment, morphology, EQE, stability, efficiency roll-off

## Abstract

We investigated the performance of single-structured light-emitting electrochemical cell (LEC) devices with Ru(bpy)_3_(PF_6_)_2_ polymer composite as an emission layer by controlling thickness and heat treatment. When the thickness was smaller than 120–150 nm, the device performance decreased because of the low optical properties and non-dense surface properties. On the other hand, when the thickness was over than 150 nm, the device had too high surface roughness, resulting in high-efficiency roll-off and poor device stability. With 150 nm thickness, the absorbance increased, and the surface roughness was low and dense, resulting in increased device characteristics and better stability. The heat treatment effect further improved the surface properties, thus improving the device characteristics. In particular, the external quantum efficiency (EQE) reduction rate was shallow at 100 °C, which indicates that the LEC device has stable operating characteristics. The LEC device exhibited a maximum luminance of 3532 cd/m^2^ and an EQE of 1.14% under 150 nm thickness and 100 °C heat treatment.

## 1. Introduction

Organic light-emitting diode (OLED) technology is used widely in highly energy-efficient displays and has always attracted worldwide attention. However, there are many issues to find a solution to the high manufacturing cost and long processing time [1,2]. The fabrication process requires the precise control of multi-layers such as the hole injection layer (HIL), the hole transport layer (HTL), the emitting layer (EML), the electron transport layer (ETL), and the electron injection layer (HIL) for high-resolution and large-area display, which complications finding solutions to the issue of high cost and manufacturing time. Therefore, it is necessary to propose a simple structure for light-emitting displays. A simple light-emitting device structure was first reported by A.J. Heeger’s group based on a luminescent conjugated polymer, to which light-emitting electrochemical cell (LEC) can be applied as an organic semiconductor [3]. The advantage of LEC devices is that they can produce excellent light-emitting properties without a charge injection and transport layer, and it is possible to reduce the manufacturing cost and processing time. Therefore, LEC displays will be easily applied as a large-area displays with a simple structure in the future.

Performance improvement research regarding changes in the composition ratio of emission layer materials or additional transport layers has continuously been conducted until recently. Adding a transport layer can reduce the loss of exciton by controlling the movement of the charge during the recombination of a charge that has been injected through the two electrodes. Moreover, a change in the synthesis ratio of the emission layer changes the number of ions produced, helping to transport charge, showing a tendency for LEC to increase its performance [4,5,6,7,8]. Among these studies, some researchers have reported on LEC devices based on iridium compounds or ruthenium compounds such as complex compounds that mix Ru(bpy)_3_^2+^ with 2,2′-bipyridyl ligands [9,10]. Tris(2,2′-bipyridine)ruthenium(II) hexafluorophosphate [Ru(bpy)_3_(PF_6_)_2_] is a compound of Ru(bpy)_3_^2+^ and two hexafluophospate [2(PF_6_)^−^]. The characteristic red light of the compound is emitted by the ions returning to the ground state through oxidation and the reduction of reactions from the following annihilation path reaction mechanisms in Ru(bpy)_3_^2+^ [11,12]:Ru(bpy)_3_^2+^ + e^−^ → Ru(bpy)^3+^(1)
Ru(bpy)_3_^2+^ − e^−^ → Ru(bpy)_3_^3+^(2)
Ru(bpy)_3_^+^ + Ru(bpy)_3_^3+^ → Ru(bpy)_3_^2+^*+ Ru(bpy)_3_^2+^(3)
Ru(bpy)_3_^2+^* → Ru(bpy)_3_^2+^ + hv(4)

However, the reaction of Ru(bpy)_3_(PF_6_)_2_ occurs in liquid, which is challenging for use in our LEC devices composed of solids during fabrication. Therefore, Ru(bpy)_3_(PF_6_)_2_ uses a luminescent material and 1-Ethyl-3-methylimidazolium bis(trifluoromethylsulfonyl)imide ([EMIM][TFSI]) to separate positive ions and negative ions, which helps transportation of the injected excitons in the process from the electrode meeting and their recombination in the emission layer to prevent losses.

Inkjet printing technology is evaluating a suitable future use of single-layer devices [13,14,15,16,17]. Although thermal evaporation technology uses more energy to stack OLED thin films, it is regrettable that it this not enough to evaporate soluble materials such as polymer, quantum dot, among others. In order to use soluble polymer materials, we need the appropriate technology and need to consider having a pixelated pattern. Therefore, inkjet printing technology is suitable because it can stack thin films with pixelated pattern through the solution discharge [18]. LEC has a suitable structure for applying inkjet printing technology [19]. Because of a single layer that prevents the need for multiple processes and because using polymers creates significant visible advantage when used in the emission layer as a material, this method takes less time and costs less as well as provides a wide range of materials [20]. In addition, if future displays evolve into stretchable and rollable forms, further research on the structural simplification of light-emitting displays will be necessary [21,22].

Currently, many researchers have continued to focus on simple-structure LEC devices [22,23,24,25]. However, the focus is on synthesizing the light-emitting material or the structural study of inserting an additional charge transport layer, and there is no study on the fabrication process of a simple structure based on a single layer. Therefore, in this paper, an optimization study on the thickness and heat treatment of the active layer using Ru(bpy)_3_(PF_6_)_2_ polymer composite in a single-structure LEC was conducted. In particular, the effect of thickness and heat treatment control on the surface morphology of the single layer was analyzed, and the relationship between the performance of the LEC device was analyzed as well.

## 2. Materials and Methods

### 2.1. Materials

We purchased Ru(bpy)_3_(PF_6_)_2_, [EMIM][TFSI], and poly(methyl methacrylate) (PMMA) from (Sigma-Aldrich Chemical Co. Seoul, Korea). Because Ru(bpy)_3_(PF_6_)_2_ and [EMIM][TFSI] may contain many impurities, we proceeded with a simple 3-stage recrystallization process (yield 44%), which consisted of substitution, purification, and the repetition of the previous two steps until a blended solution was produced. The solution material of the emission layer was blended as Ru(bpy)_3_(PF_6_)_2_:[EMIM][TFSI]:PMMA = 9:1:1 (wt%) in 5 mL of acetonitrile and was mixed for overnight.

### 2.2. Fabrication and Characterization

Figure 1 shows the structure of the LEC device (ITO/emission layer/Ag) and the material used in the emission layer. The patterned indium tin oxide (ITO) substrates were cleaned using isopropyl alcohol, acetone, and deionized water sequentially in an ultrasonic bath every 15 min. The cleaned substrates were then dried in an oven at 100 °C for 3 h. The solution material of the emission layer was spin coated on an ITO substrate in a glovebox filled with N_2_ gas. The thickness change was conducted by controlling the spin coating speed. When the spin coating speed was 250, 500, 1000, and 2000 RPM, the thickness of the thin film was 260, 150, 120, and 100 nm. After that, Ag with a thickness of 100 nm was deposited under high vacuum conditions (5 × 10^−6^ Torr). As shown Figure 1a, our LEC device used electrodes and emission layers without injection or transport layers.

The device characteristics of the device were measured using the Current-Voltage-Luminance (IVL) measurement system (PR-655 and Keithley 2400, LMS, Anyang-si, Korea). The size of the active area in the device was 2.25 mm^2^. The morphological properties were analyzed from the shape and crystal distribution on the surface using an atomic force microscope (XE-100). The UV–Vis spectrum was established at a wavelength of 200–600nm using UV–Visible spectroscopy (Evolution 600, Thermo Scientific, Seoul, Korea).

## 3. Results and Discussion

Figure 2 shows the device performance of the LEC ((a) current density–voltage (J-V), (b) luminance, (c) spectrum, and (d) external quantum efficiency (EQE)) with different emission layer thicknesses, and detailed performance parameters are specified in Table 1. The heat treatment temperature for all devices was 80 °C. As shown in Figure 2a, the turn-on voltage shows 3.4 V, 4.6 V, 4.2 V, 5.2 V at 100 nm, 120 nm, 150 nm, and 260 nm thickness, respectively. The turn-on voltage refers to the point at which the luminance increases rapidly. The turn-on voltage is also called the driving voltage or the operating voltage. This means that voltage is required to operate the display device. Overall, the turn-on voltage tends to increase as the thickness increases, but the turn-on voltage of the 150 nm device was lower than that of the 120 nm device. Moreover, the thickness of the 150 nm emission layer had a higher maximum current density than that of the other devices. It can be seen that an irregular increase regardless of the emission layer thickness can be determined from the current–voltage relationship in the device with a double carrier. When the emission layer formed a thin film, the ion involved in the charge transfer is activated and follows the relational formula below [26]:(5)$$J=\varepsilon \mu \frac{1}{d^{2}\lambda_{s}}\left(\frac{V^{2}}{2}-\frac{Vk_{B}T}{e}\ln\frac{2e\lambda_{s}V}{dk_{B}T}\right)$$*ε* is the dielectric constant, *μ* is the mobility, and *V* is the applied voltage; *k_B_* is the Boltzmann constant, *T* is the temperature, *e* is the charge of excitons, *d* is the thickness of the emission layer, and *J* is the current density. The *λ_s_* is defined as Equation (6):(6)$$\lambda_{s}=\sqrt{\frac{\varepsilon k_{B}T}{2e^{2}n_{0}}}$$

*n*_0_ is the charge density of the electrode. In this experiment, materials and processes are identical with the exception of the thickness of the emission layer. If we remove the parameters with the same values in the devices using the same material, we can see that *J* is proportional to 1/*d*^2^ at the same voltage. However, our results show an irregular tendency. In particular, the thickness of the emission layer shows the maximum value of current density at 150 nm (2251 mA/cm^2^ at 5.4 V). In the emission layer, the distribution rate of the edge ions varies in thickness, resulting in a change in the mobility for excitons. As such, it can be assumed that the relative dielectric constant has changed, affecting the current density and having the highest mobility in at the thickness of 150 nm. Moreover, this resulted in the turn-on voltage of the 150 nm device being lower than that of the 120 nm device. Figure 2b shows the luminance characteristics according to the voltage of the LEC devices with different thicknesses. The LEC devices have the characteristic of increasing luminance at a constant current/voltage [8]. Our LEC device has the characteristic that the luminance reaches the maximum value after 8 s at constant voltage for all devices and maintains constant luminance. The result of increasing the luminance at a constant voltage is the needed time for ions to react with electrons and holes. However, the time required for the ions to react may depend on the material used or the structure of the device [27,28]. In order to reduce the change in luminance due to the ion reaction in the LEC device, we measured luminance after a waiting time of 12 s after applying a voltage. Figure 2b shows that the device with a thickness of 260 nm and the 100 nm device had a luminance of 1147 cd/m^2^ and 2223 cd/m^2^, respectively. However, the devices with thicknesses of 150 nm and 120 nm had a high luminance of 3505 cd/m^2^ and 2755 cd/m^2^, respectively. This result shows that the thickness of the emission layer has maximum luminance at ~150 nm, whereas the luminance gradually decreases at too high or too low a thickness. The spectrum intensity in Figure 2c shows a similar tendency in the luminance results. The spectrum shows wavelengths from 616–648 nm for all devices. According to the thickness, the LEC device also has an effect of the microcavity effect [29]. Figure 2d shows the normalized electroluminescence (EL) spectrum of the LEC device with different thicknesses, in which the peak position of the normalized EL spectrum is redshifted with increasing thickness. In particular, a large redshift occurs in the device with a 260 nm thickness, where the thickness is significantly increased. However, where the thickness is slightly increased, a slight redshift occurs. Moreover, in the case of the devices with thicknesses of 120 nm and 150 nm, there is no significant difference. Figure 2e shows the International Commission on Illumination (or Commission International de l’Eclairage, CIE) 1931 color coordinate characteristics of LEC devices with different thicknesses. As the thickness increases, the x-coordinate increases, and the y-coordinate decreases, showing a deep red light emission characteristic. Therefore, it can be seen that spectrum changed as a result of the effect of microcavities, according to changes in the thickness of the LEC device, and thus, the emission color was also affected. The device emitted red light, and the color coordinates are (0.620, 0.379), (0.638, 0.362), (0.634, 0.366), (0.649, 0.351) for 100 nm, 120 nm, 150 nm and 260 nm, respectively. Figure 2f shows the EQE of the LEC devices with emission layer different thicknesses. The maximum efficiencies of devices with thicknesses of 100 nm, 120 nm, 150 nm, and 260 nm are 0.31, 1.20, 1.36, and 1.47%, respectively. The device with a thickness of 260 nm has the highest maximum efficiency, but the efficiency is rapidly reduced. Overall, efficiency roll-off occurs in all devices, but the 150 nm device operates stably because the EQE reduction rate is less than that of the other devices. The EQE reduction rates were 35, 26, 20, 81% for 100 nm, 120 nm, 150 nm, and 260 nm, respectively. The EQE reduction rate refers to the reduction rate of the EQE value when luminance of 1000 cd/m^2^ is increased based on the EQE value of the luminance of the maximum EQE. This means that a LEC device with a single-layer structure without a charge injection and transport layer requires optimal thickness control for high efficiency and stable operation. Therefore, 150 nm, which exhibits high efficiency and luminance and has a small efficiency roll-off, is the optimal thickness for device operation.

Figure 3 shows the morphological properties of the devices with different thicknesses through atomic force microscopy (AFM), and the inset figure represents the phase image. Table 1 shows the root mean square values (RMS; Rq). The RMS measures the average height deviations of the mean line. RMS is calculated by measuring the height of the peaks and valleys of microscopic surfaces, squaring those measurements, determining the average of the squares, and finding the square root of that number. Overall, the phase image of the inset figure indicates that dark spots are deep valleys rather than pinholes. Figure 3a shows that the surface of the 260 nm device is very rough with a RMS of 4.516 nm. Figure 3b–d shows a relatively flat surface with a roughness of 1.36, 1.35, and 1.03 nm with a thickness of 150, 120, 100 nm, respectively. The high surface roughness of the 260 nm thin film is an obstacle for the recombination of electric charges, so the device stability is poor, as shown in Figure 2d. Moreover, the 100 nm thin film showing the lowest surface roughness depicts many deep valleys on the surface because the inter-particle arrangement inside the active layer is not dense, which leads to a decrease in device efficiency. On the other hand, the thin film with a 150 nm thickness has relatively few deep valleys on the surface, and the active internal particles are densely formed. As a result, the improvement of the morphological characteristics was attributed to improved device performance.

Next, we fabricated the devices with the thickness of the emission layer fixed to 150 nm and controlled the heat treatment temperature, where we investigated how temperature affected the devices with the single-layer structure. Figure 4 shows the LEC device performance with different emission layer heat treatment temperatures, and detailed performance parameters are specified in Table 2. As shown in Figure 4a, the turn-on voltage is the same at 4.4 V for all devices, but the device with annealing at 100 °C shows a lower current density than the other devices. The devices with different heat treatment temperatures do not change to 150 nm in thickness. We draw attention to the fact that the luminance and spectrum intensity of Figure 4b and c show relatively high values at a heat treatment temperature of 100 °C. According to other research, the intensity of the thin film decreases as the heat treatment temperature increases [30]. However, since we controlled the heat treatment temperature in intervals of 20 °C, the intensity of the thin film would not have changed significantly. However, if the heat treatment temperature is increased, the surface of the thin film becomes smoother, which is due to the improvement in the luminance and the spectral intensity. Figure 4d shows the EQE characteristics of the LEC devices, and the maximum EQE shows similar values according to the heat treatment temperature. However, the efficiency roll-off characteristics are better in the 100 °C annealing device than those of other devices. The EQE reduction rates are 34%, 30%, 18% for 60 °C, 80 °C, and 100 °C, respectively. As the heat treatment temperature increases, the EQE reduction rate decreases, and this result means that the optimum heat treatment temperature is essential for stable LEC operation.

Figure 5 shows the results of the AFM measurements of the surface of the emission layer with different heat treatment temperatures. According to the results of previous studies, it has been reported that the particles on the surface of the thin film are rearranged through heat treatment [31]. We have obtained the results by keeping the temperature difference between each thin film at 20 °C. Figure 5a,b have similar roughness, which can be measured to be 1.66–1.69 nm, but Figure 5c shows some differences in a roughness, which can be measured to be 1.28 nm. Therefore, when the heat treatment temperature is increased, the surface roughness is reduced, and this result was attributed to the improved luminance and stable operation of the heat treatment device at 100 °C.

Figure 6 shows the absorption spectra of the emission layer and analyzes how it affected the performance of the LEC devices. Figure 6a shows the variation in the absorption of thin films with different thicknesses. The absorption intensity of all of the thin films increased in the spectrum range of 270 to 310 nm. According to The Beer–Lambert law [32], the absorption of the thin films is based on Equation (7).
(7)A=ειc

*A* is the absorption, *ε* is the absorption coefficient of the Ru(bpy)_3_(PF_6_)_2_, ι is the movement distance of light in the thin film, and *c* is the concentration inside the thin film. Since the variety and ratio of materials used in the thin films are all the same, *ε* and *c* have the same value. Therefore, we consider parameter ι. The results of the change in absorption can be found in Figure 6a, and we observe that the thicker the thickness, the greater the absorption intensity. Since the OLED device is a mechanism for emitting light, the absorbance of the light-emitting layer has no significant effect on device performance. However, we found that the absorption rate of the LEC devices changed with thickness, which may affect the generation of the ions through reactions with Ru(bpy)_3_^2+^ in the emission layer to create ions. Therefore, since the high absorption characteristic was shown at 150 nm thickness, the luminance and efficiency of the LEC device was also shown to be in the range of the maximum characteristics. However, in the case of a thickness of 260 nm, the surface roughness was too high despite the high absorbance characteristics, resulting in the poor stability of the device performance. Figure 6b shows the absorption spectra according to the different heat treatment temperatures of the emission layer. The absorbance is similar according to the heat treatment temperature, which does not affect the formation of ions according to the reaction. Therefore, heat treatment affects the performance of the LEC device by changing the surface properties rather than the absorbance of the thin film. Figure 6c shows Ru(bpy)_3_(PF6)_2_ polymer composite absorption and EL spectrum. The absorption spectrum of the Ru(bpy)_3_(PF6)_2_ polymer composite is 200–500 nm, but the emission spectrum is 500–800 nm. In general, the absorption and the EL spectrum of light-emitting material are not the same [25], and the emission spectrum is formed in a region that matches the emission color. Therefore, Ru(bpy)_3_(PF6)_2_ has an absorption region at a short wavelength but has an emission region at a long wavelength because it is a red light-emitting material of the color coordinates (0.634, 0.366).

## 4. Conclusions

We investigated the correlation between luminescence characteristics and morphological and optical properties of LEC devices according to the ITO/ emission layer/Ag structure with changes in the thickness and heat treatment temperatures. If the thickness was smaller than 120–150 nm, device performance decreased because of the low optical properties and non-dense surface properties. On the other hand, the 260 nm thick thin film had too much surface roughness, which interfered with the recombination of the electric charges, resulting in efficiency roll-off and poor device stability. As a result, through optimal surface properties at a thickness of 150 nm, the inner particles are arranged densely, which has a more significant impact on the transportation of excitons in ions and shows improved device performance. The heat treatment temperature has no significant effect on the optical properties but may improve the surface properties, thereby improving device performance. As a result, the heat treatment temperature was increased to 100 °C, and the surface of the thin film became smoother, which was due to the improvement of device stability. The surface and optical properties of the thin film according to the thickness and temperature control of the Ru(bpy)3(PF6)_2_ polymer composite were analyzed, and the correlation between the characteristics of the thin film and the device performance of the LEC devices were analyzed. The LEC device exhibited a maximum luminance of 3532 cd/m^2^ and an EQE of 1.14% with a 150 nm emission layer thickness and 100 °C heat treatment. In particular, the EQE reduction rate was 18%, indicating stable device operation.

## Figures and Tables

**Figure 1 micromachines-12-00843-f001:**
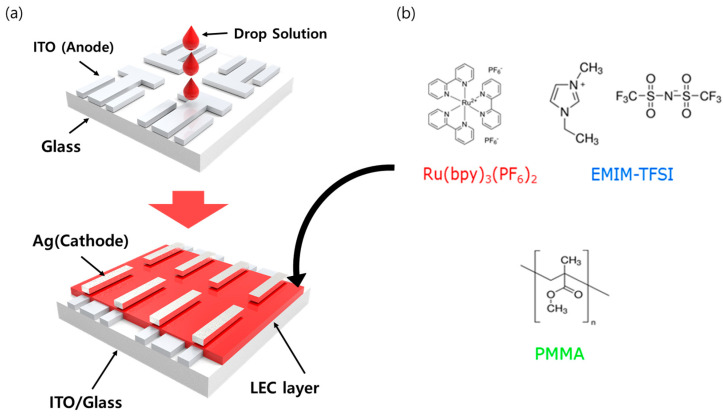
(**a**) Fabrication process of light-emitting electrochemical cell (LEC) device; Ru(bpy)_3_(PF_6_)_2_ polymer composite through a spin coating process, and Ag cathode was deposited. (**b**) Materials of the polymer composite used in the emission layer.

**Figure 2 micromachines-12-00843-f002:**
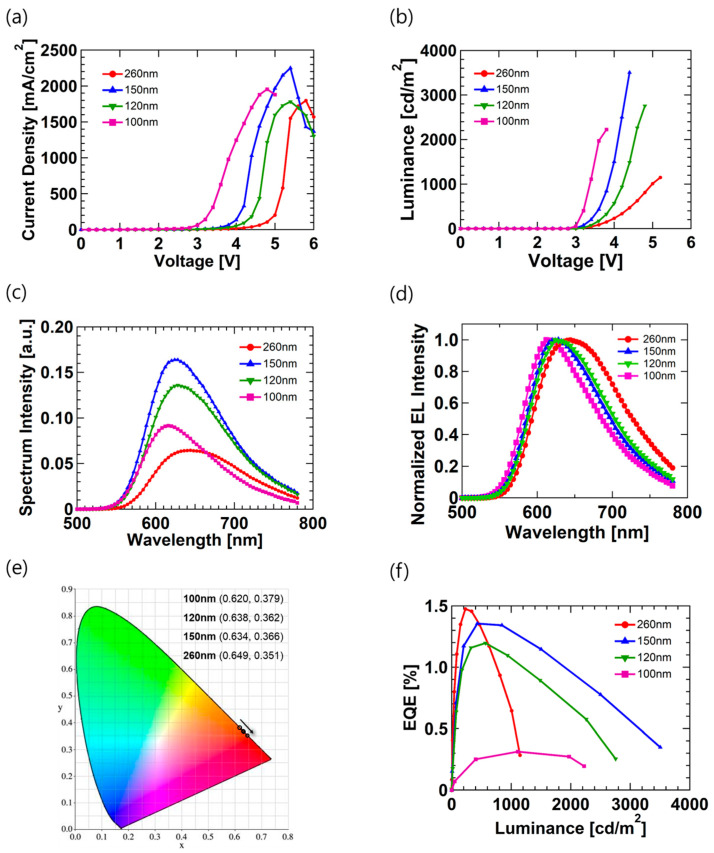
Performance of LEC devices fabricated with different emission layer thicknesses: (**a**) current density–voltage characteristics, (**b**) luminance–voltage characteristics, (**c**) spectrum intensity at wavelength, (**d**) normalized electroluminescence (EL) spectrum, (**e**) International Commission on Illumination (or Commission International de l’Eclairage, CIE) 1931 color coordinates, and (**f**) external quantum efficiency (EQE)–luminance characteristics.

**Figure 3 micromachines-12-00843-f003:**
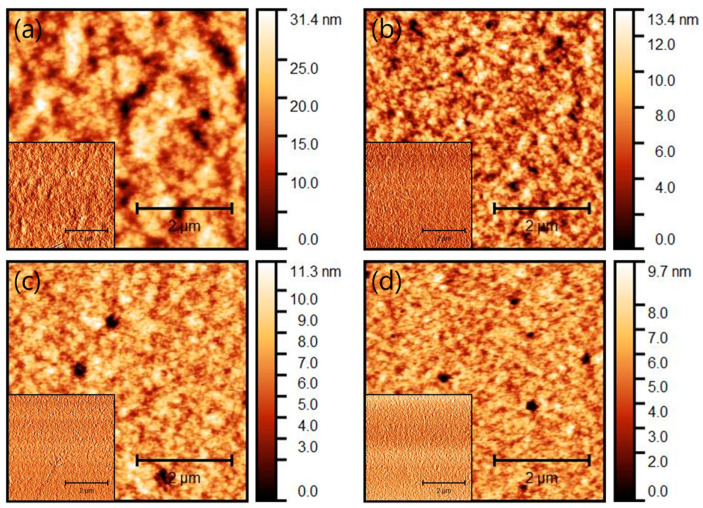
Atomic force microscopy (AFM) images of (**a**) 260 nm, (**b**) 150 nm, (**c**) 120 nm, and (**d**) 100 nm films with a scan size of 5 μm × 5 μm. (inset:phase image).

**Figure 4 micromachines-12-00843-f004:**
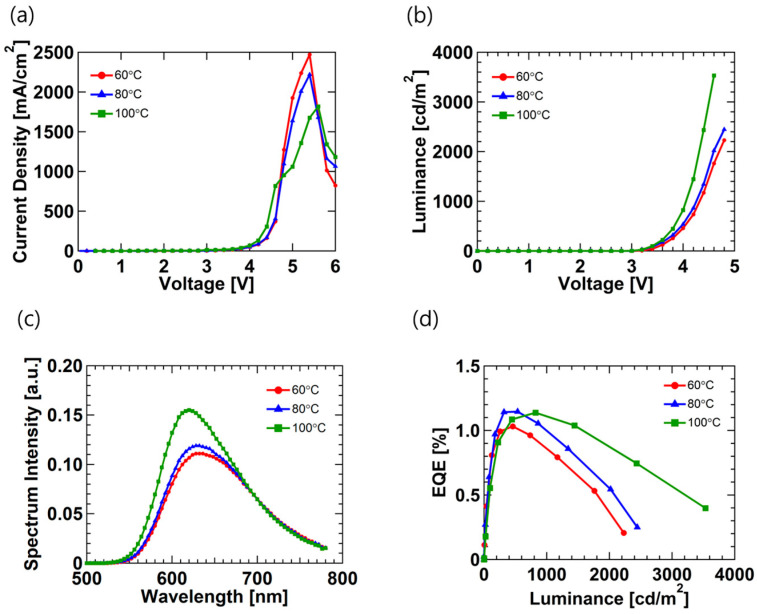
Performance of LEC devices fabricated with heat treatment of the emission layer at different temperatures:(**a**) current density–voltage characteristics, (**b**) luminance–voltage characteristics, (**c**) spectrum intensity at wavelength, and (**d**) EQE-luminance characteristics.

**Figure 5 micromachines-12-00843-f005:**

AFM images of (**a**) 60 °C, (**b**) 80 °C, and (**c**) 100 °C with a scan size of 5 μm × 5 μm.

**Figure 6 micromachines-12-00843-f006:**
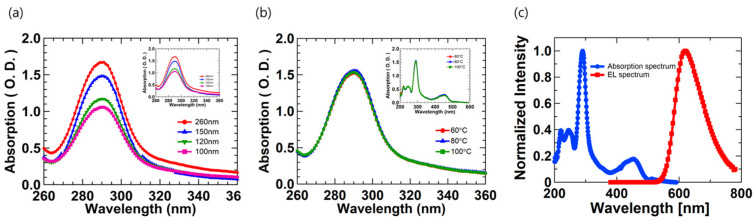
Absorption spectra at 260~360nm wavelength. (**a**) Different thicknesses of emission layers; (**b**) different emission layer heat treatment temperatures; (inset: Absorption spectra at full wavelength); (**c**) comparison of absorption (blue line) and EL spectra (red line) of Ru(bpy)_3_(PF6)_2_ polymer composite.

**Table 1 micromachines-12-00843-t001:** Luminescence characteristics and surface roughness of devices fabricated with different thickness of the emission layer.

Thickness Condition	J (mA/cm^2^)	Turn-on Voltage (V)	Luminance (cd/m^2^)	Maximum EQE (%)	RMS; Rq (nm)
260 nm	1800	5.2	1147	1.47	4.52
150 nm	2250	4.2	3505	1.36	1.36
120 nm	1782	4.6	2755	1.2	1.35
100 nm	1955	3.4	2223	0.31	1.03

**Table 2 micromachines-12-00843-t002:** Luminescence characteristics and surface roughness of devices fabricated with different emission layer heat treatment temperatures.

Temperature Condition	J (mA/cm^2^)	Turn-on Voltage (V)	Luminance(cd/m^2^)	Maximum EQE (%)	RMS; Rq (nm)
60 °C	2471	4.4	2229	1.03	1.69
80 °C	2219	4.4	2445	1.14	1.66
100 °C	1818	4.4	3532	1.14	1.28

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
