# Peer review of "Effect of Optical and Morphological Control of Single-Structured LEC Device"

_micromachines, 2021, doi:10.3390/mi12070843_

Round 1

Reviewer 1 Report

 Authors report light-emitting electrochemical cell devices with Ru(bpy)3(PF6)2 polymer composite as an emission layer. The effect of heat treatment and layer thickness was investigated using electro-optical methods and morphology study. However, the manuscript has several issues that should be fixed before any discussion on manuscript acceptance.

(1) Analysis of electrical properties.

The authors applied Mott-Gurney law to analyze the electrical properties of the sandwich device. However, the theory is applicable for single-carrier case only, and the analysis should be done using the two-carrier theory (such as plasma injected into insulators).

(2) The morphology study.

The morphology images of AFM are useless if the Authors do not plot the height scale. Moreover, authors cannot recognize if the dark spots are pinholes or not. I also recommend to include the phase images where the composite phase separation can be observed.

(3) Optical study.

The absorption is the physical process, the optical property representing the adoption process is the absorbance. Furthermore, I do not understand why the emission spectra are recorded from 500 to 780 nm, whereas the absorbance spectra are recorded from 260 to 600 nm. Therefore, there is almost no overlap, and the absorbance should also be evaluated for the emission region.

In conclusion, the results are not bad, but the concussion and not fully supported by the results. Therefore, I strongly recommend improving the analysis.

Author Response

Dear Micromachines Editor and Reviwer

The authors appreciate your e-mail and review on June 29, 2021 regarding our manuscript together with two reviewer reports. We are pleased to learn that our paper is interesting, well described for publication in Micromachines after major revisions. The valuable comments of two referees and the editor help tremendously to make the manuscript clear and competitive. We modified our paper to comply with all the reviewer comments. The revised text is expressed in yellow in the revised manuscript.

Reviewer 1)

Authors report light-emitting electrochemical cell devices with Ru(bpy)3(PF6)2 polymer composite as an emission layer. The effect of heat treatment and layer thickness was investigated using electro-optical methods and morphology study. However, the manuscript has several issues that should be fixed before any discussion on manuscript acceptance.

  1. Analysis of electrical properties. The authors applied Mott-Gurney law to analyze the electrical properties of the sandwich device. However, the theory is applicable for single-carrier case only, and the analysis should be done using the two-carrier theory (such as plasma injected into insulators).

- (Author Response) : We agree with the reviewers' good points. In the case of the Mott-Gurney law, it is widely used in single-carrier only devices, and it is not easy to use in two-carrier devices. According to the Feicht et al. paper (Proc. R. Soc. A. 2013, 469, 20130263), when a double carrier is used in OLED, electron and hole mobility and electric field should be considered, and additional parameters should be used accordingly. Therefore, we revised the manuscript by using a formula suitable for a two-carrier device.

(5); (6)

The above equation can be used in the LEC device because it is greatly affected by drift through current injection rather than diffusion. As a result, it can be seen from the above equation that the J is proportional to 1/d2. Therefore, the manuscript was revised with additional explanations, and one reference was added.

- (Revised Manuscript) Page 3~4 line 120-131: “It can be seen that an irregular increase regardless of the emission layer thickness can be guessed from the current-voltage relationship at the device with a double carrier. When the emission layer formed a thin film, the ion involved in the transfer of charge occurs and follows the relational formula below [26].

ε is the dielectric constant, μ is the mobility, and V is the applied voltage, kB is the Boltzmann constant, T is the temperature, e is the charge of excitons, d is the thickness of the emission layer, and J is the current density. The λs is defined as formula (6):

n0 is the charge density of the electrode. In this experiment, materials and processes are identical without the thickness of the emission layer. If we remove the parameter with the same value in the device using the same material, we can see that the J is proportional to 1/d2 at the same voltage.”

- (Add Reference) Page 10, line 356-357: “[26] Feicht, S.E.; Schnitzer, O.; Khair, A.S. Asymptotic analysis of double-carrier, space-charge-limited transport in organic light-emitting diodes. Proc. R. Soc. A. 2013, 469, 20130263.”

  1. The morphology study. The morphology images of AFM are useless if the Authors do not plot the height scale. Moreover, authors cannot recognize if the dark spots are pinholes or not. I also recommend to include the phase images where the composite phase separation can be observed.

- (Author Response) : We agree with the reviewer's comments. First, we modified Figure 3 and Figure 5 by presenting the height scale of the AFM image. Second, we added phase images as an inset in Figure 3, and the phase image indicates that dark spots are deep valleys rather than pinholes. As the reviewer points out, we revised the manuscript with more explanation.

- (Revised Manuscript) Page 5, line 174-180: “Figure 3 shows the morphological property of the devices with different thicknesses through the Atomic Force Microscopy (AFM), and the inset figure represents the phase image. Table 1 shows the values (RMS; Rq) of surface roughness. The RMS measures the average height deviations of the mean line. RMS is calculated by measuring the height of surfaces’ microscopic peaks and valleys, squaring those measurements, determining the average of the squares, and finding the square root of that number. Overall, the phase image of the inset figure indicates that dark spots are deep valleys rather than pinholes.”

- (Revised Figure) Page 6, line 193: “Figure 3. AFM images of (a) 260 nm (b) 150 nm (c) 120 nm (d) 100 nm films with a scan size of 5μm×5μm. (inset : phase image)”

- (Revised Figure) Page 7, line 232: “Figure 5”

  1. Optical study. The absorption is the physical process, the optical property representing the adoption process is the absorbance. Furthermore, I do not understand why the emission spectra are recorded from 500 to 780 nm, whereas the absorbance spectra are recorded from 260 to 600 nm. Therefore, there is almost no overlap, and the absorbance should also be evaluated for the emission region

- (Author Response) : Thank you very much for the reviewer's feedback. The absorption and emission spectrum of light-emitting material are generally not the same. It has already been reported in many papers. For example, reference [25] (ACS Appl. Mater. Interfaces. 2020, 12, 14254-14264) clearly shows a difference between the absorption and the emission spectrum of complex YIr in MeCN solution. Figure R1(a) is a comparison of the absorption and emission spectrum of YIr reported in the reference [25], and Figure R1 (b) is a comparison between the absorption and the emission spectrum of Ru(bpy)3(PF6)2 used in this study. The manuscript was revised with additional explanations.

[Figure R1. (a) absorption (black) and EL spectra (blue) of complex YIr in MeCN solution (b) Absorption (blue) and EL spectra (red) of Ru(bpy)3(PF6)2]

- (Revised Manuscript) Page 8, line 258-265: “Figure 6c shows the Ru(bpy)3(PF6)2 polymer composite absorption and EL spectrum. The absorption spectrum of the Ru(bpy)3(PF6)2 polymer composite is 200~500nm, but the emission spectrum is 550~800nm. In general, the absorption and EL spectrum of light-emitting material are not the same [25], and the emission spectrum is formed in a region that matches the emission color. Therefore, Ru(bpy)3(PF6)2 has an absorption region at a short wavelength but has an emission region at a long wavelength because it is a red light-emitting material of the color coordinates (0.63, 0.37).”

- (Add Figure) Page 8, line 267: “Figure 6c, (c) Comparison of absorption (blue line) and EL spectra (red line) of Ru(bpy)3(PF6)2 polymer composite.”

We think that our revised manuscript reflecting all the reviewer comments/critics is appropriate for publication in : Micromachines.

Your kind attention is sincerely appreciated.

Reviewer 2 Report

The authors reported the relationship between the device performance and the thickness/morphology of the emissive layer by analyzing morphological and optical properties. It is very useful to optimize LEC performance, but some key issues must be revised after considering the publication in the journal:

  1. In the introduction, the authors wrote a paragraph to describe the technology of inkjet printing for the advantage in LEC devices. However, it was surprising that the author did no use the technology in the manuscript.
  2. The authors utilized the different thicknesses of the emissive layer, but it should be stated that how to fabricate the different thicknesses by changing ink concentration or fabrication parameters.
  3. The authors discussed the turn-on voltage, but it should be defined.
  4. What is the difference between ‘Roughness Rs’ and ‘RMS Rq’ in Tables 1 and 2?
  5. It is hard to understand why the emissive layers with 150 nm and 260 nm show a similar absorption. It seems not to follow the beer-lambert law.
  6. Figure 1 on page 7 should be Figure 5.
  7. It is possible to discuss the effect of microcavity in the different thickness layers.
  8. LEC comprising the mobile ions in the emissive layer features a distinctive operating characteristic. The authors must show the fabricated LEC can exhibit such characteristics, e.g., increasing luminance at a constant current/voltage driving model.

Author Response

Dear Micromachines Editor and Reviewer

The authors appreciate your e-mail and review on June 29, 2021 regarding our manuscript together with two reviewer reports. We are pleased to learn that our paper is interesting, well described for publication in Micromachines after major revisions. The valuable comments of two referees and the editor help tremendously to make the manuscript clear and competitive. We modified our paper to comply with all the reviewer comments. The revised text is expressed in yellow in the revised manuscript.

Reviewer 2)

The authors reported the relationship between the device performance and the thickness/morphology of the emissive layer by analyzing morphological and optical properties. It is very useful to optimize LEC performance, but some key issues must be revised after considering the publication in the journal:

  1. In the introduction, the authors wrote a paragraph to describe the technology of inkjet printing for the advantage in LEC devices. However, it was surprising that the author did no use the technology in the manuscript.

- (Author Response) : Thank you very much for the reviewer's feedback. In the reviewer's opinion, the inkjet printing technology was described in our manuscript, but it was not applied in this experiment. We want to emphasize that inkjet printing technology is emerging as a big issue in solution processed display research. Many researchers, including the author, used spin coating technology, but inkjet printing technology will become essential to consider pixelated patterns for large area displays in the future. When researchers are interested in our research, we wanted to inform that an attempt using inkjet printing is a promising method in the future. Therefore, although the inkjet printing experiment has not been conducted, its importance should be explained in the introduction of this paper. Furthermore, solution process-based display research should be conducted based on inkjet printing technology as future work. We revised the manuscript by adding one reference paper.

- (Add Reference) Page 10, line 339-340: “[19] Chen, Z.; Li, F.; Zeng, Q.; Yang, K.; Liu, Y.; Su, Z.; Shan, G. Inkjet-printed pixelated light-emitting electrochemical cells based on cationic Ir(III) complexes. Org. Electron. 2019, 69, 336-342”

  1. The authors utilized the different thicknesses of the emissive layer, but it should be stated that how to fabricate the different thicknesses by changing ink concentration or fabrication parameters.

- (Author Response) : Thank you very much for the reviewer's feedback. The thickness change was carried out by controlling the spin coating speed. When the spin coating speed is 250, 500, 1000, and 2000 RPM, the thickness of the thin film is 260, 150, 120, and 100 nm. As the reviewer points out, we revised the manuscript with more explanation.

- (Revised Manuscript) Page 3, line 95-97: “The thickness change was carried out by controlling the spin coating speed. When the spin coating speed is 250, 500, 1000, and 2000 RPM, the thickness of the thin film is 260, 150, 120, and 100 nm.”

  1. The authors discussed the turn-on voltage, but it should be defined.

- (Author Response) : We agree with the reviewers' good points. The turn-on voltage refers to the point at which the luminance increases rapidly. The turn-on voltage is also called driving voltage or operating voltage. That is, it means a voltage is required to operate the display device. As the reviewer points out, we revised the manuscript with more explanation.

- (Revised Manuscript) Page 3, line 114-117: “The turn-on voltage refers to the point at which the luminance increases rapidly. The turn-on voltage is also called driving voltage or operating voltage. That is, it means a voltage is required to operate the display device.”

  1. What is the difference between ‘Roughness Rs’ and ‘RMS Rq’ in Tables 1 and 2?

- (Author Response) : Thank you very much for the reviewer's feedback. First, the Roughness Rs of the manuscript is a typo in the Roughness Ra. Ra and RMS Rq are measurements of surface roughness. Both measurements are done using a profilometer though the calculations differ for Ra and RMS. Ra is the roughness average, and RMS Rq is the root mean square of a surface. Both measurements are based on the heights of peaks and valleys on the surface. However, they each use the measurement differently. One peak on a surface will affect the RMS value more so than the Ra value. In our manuscript, Roughness Ra and RMS Rq are shown in Table 1 and Table 2, but the explanation is based on RMS Rq. Therefore, Roughness Ra is unnecessary information, so in the revised manuscript, Table1 and Table2 are composed only of RMS Rq values. We revised the manuscript with more explanation.

- (Add manuscript) Page 5, line 174-180: “Figure 3 shows the morphological property of the devices with different thicknesses through the Atomic Force Microscopy (AFM), and the inset figure represents the phase image. Table 1 shows the values (RMS; Rq) of surface roughness. The RMS measures the average height deviations of the mean line. RMS is calculated by measuring the height of surfaces’ microscopic peaks and valleys, squaring those measurements, determining the average of the squares, and finding the square root of that number. Overall, the phase image of the inset figure indicates that dark spots are deep valleys rather than pinholes”

- (Revised Table 1) Page 6, line 195:

Table 1.

Thickness Condition

J (mA/cm2)

Turn-on Voltage (V)

Luminance

(cd/m2)

Maximum

EQE (%)

RMS; Rq
(nm)

260 nm

1800

5.2

1147

1.47

4.52

150 nm

2250

4.2

3505

1.36

1.36

120 nm

1782

4.6

2755

1.2

1.35

100 nm

1955

3.4

2223

0.31

1.03

- (Revised Table 1) Page 7, line 233:

Table 2.

Temperature Condition

J (mA/cm2)

Turn-on Voltage (V)

Luminance

(cd/m2)

Maximum

EQE (%)

RMS; Rq
(nm)

60℃

2471

4.4

2229

1.03

1.69

80℃

2219

4.4

2445

1.14

1.66

100℃

1818

4.4

3532

1.14

1.28

  1. It is hard to understand why the emissive layers with 150 nm and 260 nm show a similar absorption. It seems not to follow the beer-lambert law.

- (Author Response) : We agree with the reviewers' good points. So, we measured the absorption of the 150 nm thin film of Ru(bpy)3(PF6)2 polymer composite again. As a result, the absorption of the 150 nm thin film showed lower absorption than that of the 260 nm thin film. Figure 6a has been revised. Moreover, as a reviewer's opinion, we compared this result with the beer-lambert law and explained it additionally in the manuscript.

- (Add Manuscript) Page 8, line 239-251: “According to The Beer-Lambert law, the absorption of the thin films is based on formula 7.

A is the absorption, ε is the absorption coefficient of the Ru(bpy)3(PF6)2, ι is the movement distance of light in the thin film, and c is the concentration inside the thin film. Since the variety and ratio of materials used in the thin films are all the same, ε and c, have the same value. Therefore, we consider parameter ι. The results of the change in absorption can be found in Figure 6a, and we observe that the thicker the thickness, the greater the absorption intensity. Since the OLED device is a mechanism for emitting light, the absorbance of the light-emitting layer has no significant effect on device performance. However, we found that the absorption rate of LEC devices changed with thickness, it may affect the generation of ions by reacting with Ru(bpy)32+ in the emission layer to create ions. Therefore, since the high absorption characteristic was shown at 150 nm thickness, the luminance and efficiency of the LEC device also showed the maximum characteristic.”

- (Add Reference) Page 10, line 367-368: “[31] Hardesty, J.H.; Attili, B. Spectrophotometry and the Beer-Lambert Law: An Improtant Analytical Technique in Chemistry. Collin College: Collin, TX, USA. 2010.”

- (Revised Figure) Page 8, line 267: “Figure 6a”

  1. Figure 1 on page 7 should be Figure 5.

- (Author Response) : Thank you for pointing out the error in the sentence. We revised the manuscript.

- (Revised Manuscript) Page 7, line 232: “Figure 5. AFM images of (a) 60℃ (b) 80℃ (c) 100℃ with a scan size of 5μm×5μm.”

  1. It is possible to discuss the effect of microcavity in the different thickness layers.

- (Author Response) : Thank you very much for the reviewer's feedback. Many researchers used the microcavity effect in a top emission OLED device, but we did not discuss it because we fabricate the device of bottom emission. However, to respond to the reviewer’s opinion, we considered the microcavity effect in a bottom emission device. In the first case, Han et al. paper (Opt. Express. 2015, 23, 19863-19873) discussed that it is often used anisotropic material with an extensive refractive index in form nanoparticle type to change the refractive index of light. However, our LEC device has a single-layer structure so that no additional layer can change the refractive index. Second, Lin et al. paper (Phys. Chem. Chem. Phys. 2015, 17, 6956-6962) reported the microcavity effect using LEC devices using non-doped blue or red materials to maximize emissions. However, our emission layer of the LEC device is synthesized with [EMIM][TFSI] and PMMA. Therefore, it is difficult to apply the microcavity effect in the case of our single-structured LEC devices. Based on the reviewers' comments, a description of the microcavity has been added with two reference papers in the revised manuscript.

- (Add Manuscript) Page 4, line 159-168: We considered whether the thickness change of the light-emitting layer affects the microcavity effect. The microcavity effect plays an essential role due to its strong influence on the extraction of light of the devices. The Han et al. paper [27] discussed that it is often used anisotropic material with an extensive refractive index in form nanoparticle type to change the refractive index of light. However, our LEC device has a single-layer structure so that no additional layer can change the refractive index. Second, Lin et al. paper [28] reported the microcavity effect using LEC devices using non-doped blue or red materials to maximize emissions. However, our emission layer of the LEC device is synthesized with [EMIM][TFSI] and PMMA materials. Therefore, it is difficult to apply the microcavity effect in the case of our single-structured LEC devices.

- (Add Reference) Page 10, line 358-362: “[27]. Han, J.H.; Kim, D.-H.; Choi, K.C. Microcavity effect using nanoparticles to enhance the efficiency of organic light emitting diodes. Opt. Express. 2015, 23, 19863-19873; [28]. Lin, G.R.; Chen, H.R.; Shih, H.C.; Hsu, J.H.; Chang, Y.; Chiu, C.H.; Cheng, C.Y.; Yeh, Y.S.; Su, H.C.; Wong, K.T. Non-doped solid-state white light-emitting electrochemical cells employing the microcavity effect. Phys. Chem. Chem. Phys. 2015, 17, 6956-6962.”

  1. LEC comprising the mobile ions in the emissive layer features a distinctive operating characteristic. The authors must show the fabricated LEC can exhibit such characteristics, e.g., increasing luminance at a constant current/voltage driving model.

- (Author Response) : Thank you very much for the reviewer's feedback. The LEC device in this study shows a tendency to increase in luminance as the voltage increases. These results have already been shown in Figure 2b and Figure 4b. Moreover, the LEC device exhibited a maximum luminance of 3532 cd/m2 and an EQE of 1.14% under 150 nm thickness and 100℃ heat treatment. Therefore, it has been described in the manuscript that the LEC device works well as a display. The operating characteristics of LEC devices are described in the manuscript ("Page 4, line 131-148”)

We think that our revised manuscript reflecting all the reviewer comments/critics is appropriate for publication in : Micromachines.

Your kind attention is sincerely appreciated.

Round 2

Reviewer 1 Report

 The authors fixed all the issues raised by the Reviewer; hence, I do not have any other possibility but to recommend the manuscript for acceptance.

Author Response

- (Author Response) : Our manuscript was revised through excellent reviews by the reviewer. We thank the reviewer for helping to publish a great paper.

Reviewer 2 Report

I appreciate the authors can revise the manuscript following the reviewer's suggestions, except No. 7 and 8. So, I recommend that the manuscript has to be revised again after considering the acceptance. 

For question 7: the authors should discuss the microcavity effect in the bottom and single-emissive LEC device, e.g. in Scientific Reports (2018) 8: 6970.

For question 8: the authors misunderstand the comment. In order to demonstrate the dynamic operational characteristic in LECs, the devices have to be driven by a 'constant' current/voltage model. It must be carried out in a further revised manuscript. 

Author Response

I appreciate the authors can revise the manuscript following the reviewer's suggestions, except No. 7 and 8. So, I recommend that the manuscript has to be revised again after considering the acceptance.

For question 7: the authors should discuss the microcavity effect in the bottom and single-emissive LEC device, e.g. in Scientific Reports (2018) 8: 6970.

- (Author Response) : Thank you very much for the reviewer's feedback. Referring to the paper (Sci. Rep. 2018, 8, 6970), it was confirmed that our LEC devices with different thicknesses have spectral changes due to the microcavity effect. So, we analyzed the microcavity effect with the normalized EL spectrum of the LEC device. Figure 2d and Figure 2e have been newly added in the second revised manuscript. Figure 2e shows that the peak position of the normalized EL spectrum is redshift with increasing thickness. In particular, a large redshift occurs in a device with 260nm, where the thickness is significantly increased. However, where the thickness is slightly increased, a slight redshift occurs, moreover in the case of the devices with 120nm and 150nm, there is no significant difference. Figure 2e shows the CIE 1931 color coordinate characteristics of LEC devices with different thicknesses. As the thickness increases, the x-coordinate increases, and the y-coordinate decreases, showing a deep red light emission characteristic. Therefore, it can be seen that spectrum changed by microcavity effect according to change in the thickness of the LEC device, and thus, the emission color also affected. Based on the reviewers' comments, a description has been added with one reference paper in the second revised manuscript.

- (Add Manuscript) Page 4, line 152-165: “According to the thickness, the LEC device is also an effect of the microcavity effect [29]. Figure 2d shows the normalized EL spectrum of the LEC device with different thicknesses, in which the peak position of the normalized EL spectrum is redshifted with increasing thickness. In particular, a large redshift occurs in a device with 260nm, where the thickness is significantly increased. However, where the thickness is slightly increased, a slight redshift occurs, moreover in the case of the devices with 120nm and 150nm, there is no significant difference. Figure 2e shows the CIE 1931 color coordinate characteristics of LEC devices with different thicknesses. As the thickness increases, the x-coordinate increases, and the y-coordinate decreases, it was showing a deep red light emission characteristic. Therefore, it can be seen that spectrum changed by microcavity effect according to change in the thickness of the LEC device, and thus, the emission color also affected. The device emitted red light and the color coordinates are (0.620, 0.379), (0.638, 0.362), (0.634, 0.366), (0.649, 0.351) for 100nm, 120 nm, 150 nm and 260 nm respectively.”

- (Add Figure) Page 4, line 180 : Figure 2d and Figure 2

- (Add Reference) Page 10, line 373-374: “[29] Lindh, E.M.; Lundberg, P.; Lanz, T.; Mindermark, J.; Edman, L. The Weak Microcavity as an Enabler for Bright and Fault-tolerant Light-emitting Electrochemical Cells. Sci. Rep. 2018, 8, 6970.”

For question 8: the authors misunderstand the comment. In order to demonstrate the dynamic operational characteristic in LECs, the devices have to be driven by a 'constant' current/voltage model. It must be carried out in a further revised manuscript.

- (Author Response) : Thank you very much for the reviewer's feedback, and we understand the reviewer's opinion. Our LEC thin film device has the characteristic of increasing luminance at a constant voltage. Figure R1 results from measuring the increase in luminance in the constant driving model (constant voltage @ 4V). It can be confirmed that it reaches the maximum luminance after 8 seconds for all devices and maintains constant luminance. The result of increasing the luminance at a constant voltage is the time for ions to react with electrons and holes, which have been widely reported in LEC research results. However, the time required for ions to react may depend on the used material or the structure of the device. [27-28] To reduce the change in luminance due to the ion reaction of the LEC device, we measured luminance after a waiting time of 12 seconds after applying a voltage. Therefore, the data of figures 2b and 4b shows the maximum luminance value at voltage. Based on the reviewers' comments, a description has been added with two reference papers in the second revised manuscript.

- (Add Manuscript) Page 4, line 137-146: “Figure 2b shows the luminance characteristics according to the voltage of LEC devices with different thicknesses. The LEC device has a characteristic of increasing luminance at constant current/voltage [8]. Our LEC device has a characteristic that the luminance reaches the maximum value after 8 seconds at constant voltage for all devices and maintains constant luminance. The result of increasing the luminance at a constant voltage is the time for ions to react with electrons and holes. However, the time required for ions to react may depend on the used material or the structure of the device [27-28]. In order to reduce the change in luminance due to the ion reaction of the LEC device, we measured luminance after a waiting time of 12 seconds after applying a voltage.”

- (Add Reference) Page 10, line 369-372: “[27] Gorodetsky, A.A.; Parker, S.; Slinker, J.D.; Bernards, D.A.; Wong, M.H.; Malliaras, G.G.; Flores-Torres, S.; Abruña, H.D. Con-tact issues in electroluminescent devices from ruthenium complexes. Appl. Phys. Lett. 2004, 84, 807-809; [28] Buda, M.; Kalyuzhny, G.; Bard, A.J. Thin-Film Solid-state Electroluminescent Devices Based on Tris(2,2’- bipyri-dine)ruthenium(II) Complexes. J. Am. Chem. Soc. 2002, 124, 6090-6098.”
